# Anti-SARS-CoV-2 Antibody Responses 5 Months Post Complete Vaccination of Moroccan Healthcare Workers

**DOI:** 10.3390/vaccines10030465

**Published:** 2022-03-18

**Authors:** Najlaa Assaid, Soukaina Arich, Hicham Charoute, Khadija Akarid, Sayeh Ezzikouri, Abderrahmane Maaroufi, M’hammed Sarih

**Affiliations:** 1Service de Parasitologie et des Maladies Vectorielles, Institut Pasteur du Maroc, Place Louis Pasteur, Casablanca 20360, Morocco; najlaa.essaid@gmail.com (N.A.); arichsoukaina@gmail.com (S.A.); abderrahmane.maaroufi@pasteur.ma (A.M.); 2Health and Environment Laboratory, Molecular Genetics and Immunophysiopathology Research Team, Aïn Chock Faculty of Sciences, University of Hassan II Casablanca (UH2C), Casablanca 20100, Morocco; kakarid@yahoo.fr; 3Research Unit of Epidemiology, Biostatistics and Bioinformatics, Institut Pasteur du Maroc, Casablanca 20360, Morocco; hicham.charoute@pasteur.ma; 4Viral Hepatitis Laboratory, Virology Unit, Institut Pasteur du Maroc, Casablanca 20360, Morocco; sayeh.ezzikouri@pasteur.ma

**Keywords:** COVID vaccine, ChAdOx1 nCoV-19, BBIBP-CorV, antibody response

## Abstract

Data about the duration of antibodies after vaccination show that the protection against SARS-CoV-2 infection begins to decline over time. This study aims to determine anti-SARS-CoV-2 anti-S IgG levels in healthcare workers five months after the second vaccination dose. We collected samples from 82 participants who were fully vaccinated with ChAdOx1 nCoV-19 or BBIBP-CorV. We assessed anti-SARS-CoV-2 IgG antibodies using a Euroimmun ELISA and an Abbott Architect ™ SARS-CoV-2 IgG test. Of the 82 participants, 65.85% were seropositive for IgG using ELISA, and 86.59% were positive for IgG according to the Abbott Architect ™ test. Individuals vaccinated with the ChAdOx1 nCoV-19 vaccine had a median anti-S1 antibody level of 1.810 AU/mL [interquartile range (IQR), 1.080–3.7340] and 171.7 AU/mL [79.9–684.6] according to the Euroimmun ELISA and Abbott Architect test, respectively. These tests indicated that people vaccinated with BBIBP-CorV had a median anti-S1 antibody level of 1.840 AU/mL [0.810–2.960] and 126.7 AU/mL [54.9–474.3], respectively. Statistical analysis showed no significant difference between the positivity rates of the vaccinated individuals, either for gender or for age. In addition, we found no significant difference between the two vaccines. Our study provides information on the longevity of the anti-SARS-CoV-2 IgG antibodies in people at least five months after vaccination.

## 1. Introduction

Severe acute respiratory syndrome coronavirus 2 (SARS-CoV-2) emerged in December 2019 in Wuhan, China. It spread rapidly worldwide, causing a pandemic [1]. SARS-CoV-2 caused the coronavirus disease in 2019 (COVID-19). The spectrum of COVID-19 symptoms ranges from asymptomatic disease to mild respiratory issues from asymptomatic infection, to an acute respiratory distress syndrome, which can be fatal [2]. Since the declaration of the pandemic by the World Health Organization, the coronavirus disease has spread to several countries worldwide. It has infected millions of patients and caused thousands of deaths [3]. 

Since the emergence of SARS-CoV-2, assessing and understanding the kinetics and protective role of the human immune response against this virus, has been of a high priority and relevance [4]. For this purpose, different investigations have conducted several seroepidemiological surveys worldwide to measure the seroprevalence of antibodies against SARS-CoV-2 [5,6,7,8,9,10,11]. These studies have demonstrated the importance of the humoral immune response for the elimination of viral infection, and the important role of understanding the dynamics of antibodies against SARS-CoV-2, which help not only in assessing immunological levels and predicting potential immune protection, but also in the development of vaccine and immune therapy strategies [12]. 

After a natural infection caused by SARS-CoV-2, an immune response fights the infection allowing more than 90% of patients to recover naturally. However, the level of anti-SARS-CoV-2 antibodies starts to decrease after three to four months, particularly in asymptomatic subjects [13,14]. Other studies have shown that antibody levels decrease during the first six months after infection, especially in immunocompromised subjects [15,16]. Thus, vaccination remains important in managing the COVID-19 pandemic, and in building protective immunity [17]. Several vaccines have been developed to prevent SARS-CoV-2 infection, and some have been approved for use in different countries around the world [18]. ChAdOx1 nCoV-19 (Oxford University-AstraZeneca, UK) and BBIBP-CorV (Sinopharm, the Beijing Institute of Biological Products, China) are among the approved vaccines [19]. The antibody response stimulated by these vaccines is directly related to the age of individuals and their physiological status, including their immunological status [18,20].

The Moroccan Ministry of Health declared the first case of infection on 2 March 2020. Like several countries around the world, Morocco implemented measures to prevent the spread of the disease among Moroccans [21]. From January 2021, the Kingdom of Morocco started anti-COVID-19 vaccination campaigns. It gave priority to front-line workers: healthcare workers, government authorities, the security forces, those involved in the national education system, the elderly, and people vulnerable to the virus, after which COVID-19 vaccinations were made available to the general public aged 12 years and older [22].

Some studies have shown that vaccines, including BNT162b2 mRNA (Pfizer-BioNTech, Comirnaty. Pfizer, New York, NY, USA), reduce viral load and prevent infection by reducing infectivity [23,24,25,26,27]. However, other studies have shown that the vaccine-induced immune response can decline over time [28,29] with antibody levels beginning to decline at three months, and disappearing at six months after vaccination [24,30,31]. This vaccine-induced antibody level is especially reduced in people aged 65–80 years and in immunocompromised individuals [32,33,34,35]. This is of concern, given that these categories of people have an increased risk of severe disease [36]. To address waning immunity over time, some countries have considered a third dose of COVID-19 mRNA vaccine as a booster dose for those who received the second dose at least five months prior [37,38,39]. However, the associated factors and duration of antibodies induced after vaccination have yet to be studied in several countries, including Morocco. 

To date, and to our knowledge, no data describing the vaccine immune response or its durability are available in Morocco. In the present work, we assessed the levels of anti-SARS-CoV-2 IgG antibodies present in 82 healthcare workers who were fully vaccinated with ChAdOx1 nCoV-19 (COVISHIELD, AstraZeneca, Serum Institute of India Pvt Ltd., Maharashtra, India) or BBIBP-CorV inactivated virus vaccine (Sinopharm, Beijing CNBG, Beijing, China) five months after the second dose of the vaccine. 

## 2. Materials and Methods

### 2.1. Clinical Study Procedures

A total of 82 healthcare workers participated in this study. Here, we present the data from health professionals of the Moulay Youssef Regional Hospital in Casablanca, and those of the Pasteur Institute of Morocco. The National Immunization Program began in January 2021, and has prioritized health workers. All participants in this study gave their informed consent before participating. The study protocol complied with the Helsinki Declaration. The Ethics Committee of the Mohammed VI University of Health Sciences in Casablanca approved the study. 

Study participants were adults, men and women, who had received the second dose of vaccine at least five months before the enrollment date, could provide informed consent, and had not tested positive for COVID-19 at the time of enrollment. Participants received the second dose of ChAdOx1 nCoV-19 or BBIBP-CorV vaccine at the same time (March 2021). After five months (August 2021), we collected samples from participants to measure the level of anti-SARS-CoV-2 antibodies. Volunteers were reported to have never been infected with SARS-CoV-2. Under the general rules regarding data protection, the contact details of the subjects were kept confidential, and after collecting the samples, the names were deleted and replaced by patient codes. 

### 2.2. Detection of Anti-SARS-CoV-2 IgG Antibodies 

To detect IgG antibodies against SARS-CoV-2 spike protein in human serum, we used two serological techniques: a semi quantitative enzyme-linked immunosorbent assay (ELISA) using the commercial kit (Anti-SARS-CoV-2 S1 ELISA IgG; Euroimmun, Lübeck, Germany) following the manufacturer’s instructions [40]. This method has an estimated sensitivity of 75.0% (95% CI 47.62–92.73 (based on >14 days post symptoms onset) and a specificity of 96.7% (95% CI; 92.54–98.93) [41]. The results were analyzed semi-quantitatively and interpreted according to the manufacturer’s instructions. We evaluated them by calculating a ratio of extinction of the control or patient sample to the extinction of the calibrator. This ratio is interpreted as follows: <0.8 negative; ≥0.8 to <1.0 limit; 1.1 positive. The borderline results were considered negative for the analysis. A quantitative immunoassay of chemiluminescent microparticles (CMIA) to detect antibodies against the receptor-binding domain (RBD) of the S1 subunit of the SARS-CoV-2 spike protein was used. The sequence used for the RBD was from the WH-Human 1 coronavirus. The assay was performed on the Abbott Architect i2000SR instrument (Abbott Laboratories, Abbott Park, Illinois) following the manufacturer’s instructions for the SARS-CoV-2 IgG II Quant [42]. This method has a sensitivity of (based on >14 days post-positive reverse transcription-PCR (RT-PCR) samples) and specificity of 98.3% (90.6% to 100%) and 99.5% (97.1% to 100%), respectively [42]. This assay showed high concordance with the neutralizing antibody titers [43] and can identify antibodies in patients with two variants of concern (VOC 202012/V1 [UK] and (VOC 202012/V2 (South Africa) strains [42]. The analytical measurement range defined by the manufacturer was 21 to 40,000 AU/mL and the cut-off point was ≥50 AU/mL. The protective threshold value for the Abbott Architect test is ≥4000 AU/mL [44].

### 2.3. Statistical Analysis

Association analysis between categorical variables was performed using the Chi-square test if all expected values were greater than 5. Otherwise, we used the Fisher exact test. A Mann–Whitney test was applied for the comparison of quantitative data. Values less than 0.05 were considered statistically significant. All statistical analyses were performed with the R package (https://www.r-project.org accessed on 4 January 2022).

## 3. Results

### 3.1. Demographic Data

During August 2021, a total of 82 healthcare workers were tested for the detection of anti-SARS-CoV-2 IgG antibodies by two serological techniques: the Euroimmun ELISA and the Abbott Architect ™ SARS-CoV-2 IgG tests. The study population was vaccinated, had not previously tested positive for COVID-19, and had received the second vaccine dose at the same time (5 months before the blood sample was taken for the study). Of these individuals, 41 (50%) participants were vaccinated with ChAdOx1 nCoV-19 vaccine (COVISHIELD, AstraZeneca, Serum Institute of India Pvt Ltd.) (PVA) of which 48.78% were women (*n* = 20) and 51.22% were men (*n* = 21), and 41 (50%) with BBIBP-CorV inactivated virus vaccine (Sinopharm, Beijing CNBG) (PVS), of which 53.65% were women (*n* = 22) and 46.34% were men (*n* = 19). The median age of participants in this study was 54 years (range from 24 to 67 years). To check whether vaccinated individuals were previously infected with SARS-CoV-2 infection, we assessed IgG antibodies against SARS-CoV-2 nucleocapsid protein using the SARS-CoV-2 IgG test (Architect IgG test, Abbott, Chicago, IL, USA). The anti-N antibody test was performed on participants who had been vaccinated with the ChAdOx1 nCoV-19 vaccine (42). All participants were found to be negative, except one. The latter was excluded from the study (data not shown).

### 3.2. Anti S IgG Antibodies Detection

The Euroimmun ELISA and the Abbot Architect tests demonstrated an overall prevalence rate of IgG antibody seropositivity of 65.85% and 86.59%, respectively (Table 1). Women had a higher positivity rate than men (Euroimmun test: 31 (57.41%) women versus 23 (42.59%) men; Architect test: 39 (54.93%) women versus 32 (45.07%) men, but the differences were not statistically significant (*p* = 0.119 and *p* = 0.088, respectively) (Table 1). The vaccinated participants over 50 years of age had a higher positivity rate than those aged less than 50 years, but this was not significant (Euroimmun test: 38 (70.37%) versus 16 (29.63%), Architect test: 47 (66.20%) versus 24 (33.80%) with *p* = 0.231 and *p* = 1.000, respectively) (Table 1).

The Euroimmun test demonstrated that, out of 82 samples, 54 (65.85%) had anti-SARS-CoV-2 IgG antibodies, including 30 (55.56%) who were vaccinated with PVA and 24 (44.44%) with PVS (Table 2). Using the Abbott Architect test, the rates of IgG are 52.11% in PVA and 47.89% in PVS (Table 2). Out of the seropositive men, those vaccinated with the ChAdOx1 nCoV-19 vaccine represent a higher positivity rate than those vaccinated with the BBIBP-CorV vaccine, but the differences were not statistically significant (Euroimmun: 13 (56.52%) vs. 10 (43.48%) *p* = 0.554. Architect: 18 (56.25%) versus 14 (43.75%); *p* = 0.442) (Table 2). For positive women, those vaccinated with ChAdOx1 nCoV-19 represent a higher positivity rate than those vaccinated with BBIBP-CorV with a statistically non-significant difference (Euroimmun: 17 (54.84%) versus 14 (45.16%); *p* = 0.116). People over 50 years of age vaccinated with ChAdOx1 nCoV-19 have a higher positivity rate than those vaccinated with BBIBP-CorV (60.53% vs. 39.47% and 57.45% vs. 42.55%, respectively, by Euroimmun and Architect). On the other hand, people aged less than 50 years, PVS has a higher positivity rate than in PVA; but statistically, these results are not significant (Table 2).

The antibody levels in both groups (PVA group (*n* = 41) and PVS group (*n* = 41)) were analyzed (Figure 1). The Euroimmun test shows that the IgGs in the two groups were above the cutoff value (1.1) and that the antibodies titers were almost similar for the two groups (*p* = 0.409) (Figure 1B). The Abbott Architect assay shows similar results to the Euroimmun test and there is no significant difference between the level of IgG in the two groups PVA and PVS (*p* = 0.273) (Figure 1A). PVA had a median anti-S1 antibody level of 1.810 [interquartile range (IQR), 1.080–3.340] AU/mL, and 171.7 [79.9–684.6] AU/mL respectively, according to the Euroimmun and Architect assays (Figure 1). PVS had a median anti-S1 antibody level of 1.840 [IQR, 0.810–2.960] AU/mL and 126.7 [IQR, 54.9–474.3] AU/mL respectively, according to the Euroimmun and Architect tests (Figure 1).

## 4. Discussion

Vaccination against SARS-CoV-2 is a key factor in achieving protective immunity against severe forms of the disease. As of early 2021, several vaccines have been used to limit the burden of SARS-CoV-2 infection. Achieving effective and sustained immunity depends on vaccines inducing an immune response comparable to the response elicited by the virus they target. COVID-19 poses a significant health risk to healthcare workers worldwide, and several studies have demonstrated that these individuals are at high risk of infection [45,46,47,48]. As a result, these individuals were identified as priority groups for the COVID-19 vaccine allocation [49]. From January 2021, the Kingdom of Morocco launched a vaccination program against COVID-19, with AstraZeneca/Covishield produced by the Serum Institute of India in India and Sinopharm/BBIBP-CorV produced in China, and has given priority to health professionals [22]. However, several studies show that the protection against SARS-CoV-2 infection after vaccination, declines over time [24,28,29,30]. 

In the present study, we investigated the level of anti-S-IgG antibodies to SARS-CoV-2 five months after full vaccination with Covishield or BBIBP-CorV, in 82 Moroccan healthcare workers, who have never been infected with SARS-CoV2. We found that anti-S-IgG antibodies persisted for five months after the second dose but with a relatively low anti-S IgG titer for both vaccines. We reported an overall anti-S IgG antibody prevalence rate of 86.59%. Those vaccinated with Covishield/AstraZeneca had a median anti-S1 antibody level of 171.7 AU/mL [IQR, 79.9–684.6], and those vaccinated with BBIBP-CorV/Sinopharm had a median anti-S1 antibody level of 126.7 AU/mL [IQR, 54.9–474.3]. A study was conducted in India in individuals who were vaccinated with Covishield, using the same serological test that we used in this study (Abbott Architect kit), and it reported a decrease in antibody levels five months after the second dose. This is still high compared to our study (637.2 AU/mL vs. 171.7 AU/mL) [50]. This difference in antibody levels may be due, in addition to geographical and racial variations, to the small number of participants in our study compared to the Indian study (41 vs. 308, respectively). In a study conducted by Mishra and his collaborators on 122 health workers who received a full dose of ChAdOx1 nCoV-19 vaccine, a high seroprevalence rate (94.26%) similar to the prevalence obtained in our study (90%, 37/41) was reported [51]. A study conducted on 203 BBIBP-CorV vaccine recipients showed a high seroprevalence rate of 95.07% three months post-vaccination [52]. 

Various studies that were conducted in order to understand the kinetics and durability of antibodies induced after vaccination (ChAdOx1 nCoV-19 and BBIBP-CorV) showed that the immune response persists for up to three to four months, but begins to decline significantly over time [51,52,53,54,55,56,57]. A study of 122 healthy individuals vaccinated with BNT162b2, where the authors used the same test used in this study to determine the level of antibodies six months after the second dose, reported a decrease in the level of antibodies, but this level was still very high compared to our study (1383 AU/mL). This high value could be the fact that the mRNA vaccine induces high antibody levels compared to the vaccines in our study [30]. Another study conducted on COVID-19-naïve subjects from Southeast Asia vaccinated with the BNT162b2 vaccine, using the same assay showed a mean value of 1210 AU/mL five months after the second dose [58]. The mRNA vaccines, BNT162b2/Pfizer and Moderna/mRNA-1273, which are approved as the most effective vaccines, induce a strong humoral response but begin to decline six months after the second dose [59,60]. The decrease in the post-vaccine humoral response may be expected, as the vaccine-induced humoral immunological memory represented by memory plasma cells, may not be long-lived [61]. However, some data suggest the persistence of memory B cells that can, upon reinfection, produce rapid and enhanced antibodies [62,63]. Additionally, antibody levels induced after natural infection with SARS-CoV-2 persist for up to 13 months, but begin to decline with time [57,64,65]. 

In our study, statistical analysis showed no significant difference between the positivity rates of vaccinated individuals, either with gender or with age. Our results are consistent with some studies that demonstrate that there is no significant difference between the sex of the participants and the rate of seropositivity. On the other hand, there is a difference regarding the age of the participants, so the rate of seropositivity was significantly lower in participants aged over 60 years [55,66,67]. This can be explained by the fact that age is among the important factors influencing the immune response to vaccination. Some studies reported that older people reacted poorly to influenza and hepatitis vaccines [68]. Similar to our results, Choudhary and his collaborators describe no significant difference in post-vaccination antibody production according to sex and age [50]. In contrast, another study reported a high level of antibodies in young participants and women [53]. We also found that the level of IgG seropositivity in subjects vaccinated with Covishield or BBIBP-CorV was not significantly different between the two vaccines (171.7 AU/mL vs. 126.7 AU/mL, respectively) and this is probably due to sample size. The cellular immune response may be different between the two vaccines, which was not evaluated in our study. It would also be interesting to compare the humoral response after vaccination at different timepoints. Other studies showed that the Covishield vaccine produced higher anti-S IgG titer than induced by Covaxin [50,67]. 

Several studies have shown the efficacy of the ChAdOx1 nCoV-19 vaccine in the prevention of symptomatic and severe cases of COVID-19 in different populations including the elderly (significant vaccine efficacy of 70.4% after two doses) [69,70,71]. However, its efficacy, represented by neutralizing antibody activity, is much less when compared to mRNA vaccines [72]. Indeed, 90% efficacy was reported for the BNT162b2 vaccine [73] and 94.5% for the mRNA-1273 vaccine [74]. An efficacy of 78.1% in healthy adults has been shown for the BBIBP-CorV inactivated vaccine [75].

In the present study, we determined the level of anti-RBD IgG antibodies by using the Architect assay, but we were unable to determine the neutralizing antibodies that protect against infection due to the unavailability of a neutralization test and our limited resources. However, some studies have shown a correlation between tests measuring anti-RBD IgG and SARS-CoV-2 neutralizing antibody titers [18,76,77]. It has also been reported that anti-RBD antibodies are responsible for most of the neutralization activity [78,79]. Indeed, the Abbott Architect technique used in our study has an estimated sensitivity of 98.3% (95% CI, 90.6–100%) [42], much higher than the Euroimmun ELISA test (75.0%) (95% CI, 47.62–92.73%) [41]. Therefore, the use of a high-sensitivity quantitative test is recommended for obtaining reliable serological results.

The study has some limitations, since only healthcare workers were included, which limits generalizability. Antibody neutralization titers are not estimated, given our limited resources. A larger sample size involving all hospital or Pasteur Institute departments in Morocco would have better reflected seroprevalence and been more informative; providing an important basis for comparison between the two types of vaccines. However, to our knowledge, this is the first study of antibody durability in individuals who received both vaccine doses in North Africa. 

Further studies are needed to monitor post-vaccination immune responses beyond six months and the inclusion of more individuals, to determine the effect of a third dose of vaccine on the duration of vaccine effectiveness, in particular against emerging variants of SARS-CoV-2.

## Figures and Tables

**Figure 1 vaccines-10-00465-f001:**
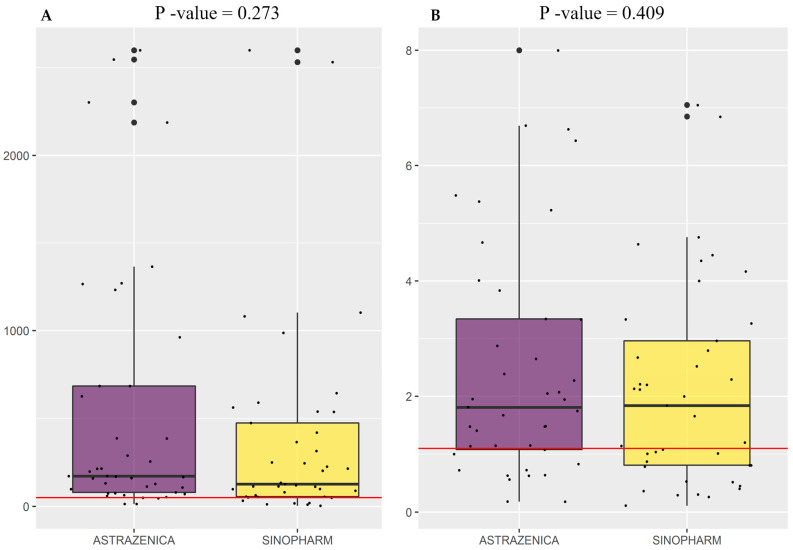
Distributions of and differences in SARS-CoV-2 IgG antibodies in ChAdOx1 nCoV-19 vaccinated participants (PVA) and BBIBP-CorV vaccinated participants (PVS) five months after the second vaccine dose. (**A**) Distributions and differences of SARS-CoV-2 IgG antibodies obtained by Abbott Architect ™ assay in ChAdOx1 nCoV-19 vaccinated participants (*n* = 41), and BBIBP-CorV vaccinated participants (*n* = 41). The boxplot displays the IgG titers of the 25th, 50th, and 75th percentiles. The red line represents the cutoff value (50). The bold line corresponds to the median value (Median = 171.7 for ChAdOx1 nCoV-19, Median = 126.7 BBIBP-CorV). (**B**) Distributions and differences in SARS-CoV-2 IgG antibodies obtained by the Euroimmun ELISA test in ChAdOx1 nCoV-19 vaccinated participants (*n* = 41), and BBIBP-CorV vaccinated participants (*n* = 41). The red line represents the cutoff value (1.1). The bold line corresponds to the median values Median = 1.810 for ChAdOx1 nCoV-19, Median = 1.840 BBIBP-CorV). *p*-value > 0.05 for the two graphs, which is not considered statistically significant (*p* = 0.273, *p* = 0.409).

**Table 1 vaccines-10-00465-t001:** Distribution of positive participants by gender and age group.

Serological Test	Euroimmun			Abbot Architect ™		
Group	Negative *n* (%)	Positive *n* (%)	*p* Value	Negative (%)	Positive *n* (%)	*p* Value
Overall	28 (34.15)	54 (65.85)		11 (13.41)	71 (86.59)	
Female	11 (39.29)	31 (57.41)	0.119	3 (27.27)	39 (54.93)	0.088
Male	17 (60.71)	23 (42.59)		8 (72.73)	32 (45.07)	
<50	12 (42.86)	16 (29.63)	0.231	4 (36.36)	24 (33.80)	1.000
>50	16 (57.14)	38 (70.37)		7 (63.64)	47 (66.20)	

**Table 2 vaccines-10-00465-t002:** EUROMMUN ELISA and Abbott Architect ™ SARS-CoV-2 IgG assay for detection of IgG antibodies.

		Euroimmun			Abbot Architect ™		
Group	Vaccine	Positive *n* (%)	Negative *n* (%)	*p*-Value	Positive *n* (%)	Negative *n* (%)	*p*-Value
Overall		54 (65.85%)	28 (34.15%)		71 (86.59%)	11 (13.41%)	
	ChAdOx1 nCoV-19	30 (55.56%)	11 (39.29%)	0.162	37 (52.11%)	4 (36.36%)	0.331
	BBIBP-CorV	24 (44.44%)	17 (60.71%)		34 (47.89%)	7 (63.64%)	
By Sex							
Male	ChAdOx1 nCoV-19	13 (56.52%)	8 (47.06%)	0.554	18 (56.25%)	3 (37.50%)	0.442
	BBIBP-CorV	10 (43.48%)	9 (52.94%)		14 (43.75%)	5 (62.50%)	
Female	ChAdOx1 nCoV-19	17 (54.84%)	3 (27.27%)	0.116	19 (48.72%)	1 (33.33%)	1.000
	BBIBP-CorV	14 (45.16%)	8 (72.73%)		20 (51.28%)	2 (66.67%)	
By Age							
<50 Years	ChAdOx1 nCoV-19	7 (43.75%)	3 (25.00%)	0.434	10 (41.67%)	0 (0%)	0.265
	BBIBP-CorV	9 (56.25%)	9 (75.00%)		14 (58.33%)	4 (100%)	
≥50 Years	ChAdOx1 nCoV-19	23 (60.53%)	8 (50.00%)	0.475	27 (57.45%)	4 (57.14%)	1.000
	BBIBP-CorV	15 (39.47%)	8 (50.00%)		20 (42.55%)	3 (42.86%)	

## Data Availability

The data presented in this study are available on request from the corresponding author. The data are not publicly available according to the ethical committee decision on the conduct of this study.

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
