# Peer review of "Anti-SARS-CoV-2 Antibody Responses 5 Months Post Complete Vaccination of Moroccan Healthcare Workers"

_vaccines, 2022, doi:10.3390/vaccines10030465_

Round 1

Reviewer 1 Report

What is the reason for comparing IgG with 2 different assays - one semi-quantitative and the other quantitative?

Please provide more details of the assays & their performance minimally from the package insert; preferably from published evaluations of these assays. Revise section 2.2 to make it clearer that all descriptions pertaining to assay (i) and (ii) is together.

Since the IgG assays are semi-quantitative/quantitative expressing the results in concentration or magnitude of cut-off index will be more useful to compare between the 2 different vaccines rather than express results as percent positive or negative. 

What is the evidence that your healthcare workers might not have been exposed to subclinical/asymptomatic CoVID19 in the 5 months after their vaccination?     

Several of your references cited are incomplete without journal title, volume, page numbers or doi - 5, 16, 28, 29, 30, 33, 35, 37, 39, 43, 45, 46, 57, 64, especially the MedRxIV citations : 23, 26, 47, 48, 50, 58, 

Author Response

Response to Reviewer 1 Comments

All modifications in the manuscript are highlighted in yellow

Point 1: What is the reason for comparing IgG with 2 different assays - one semi-quantitative and the other quantitative ?

Response 1: Thank you for your comment. Initially, the semi-quantitative EUROIMMUN ELISA was performed to assess anti-SARS-CoV-2 S1 IgG antibodies with an estimated sensitivity is 75.0% (95% CI 47.62-92.73 (based on > 14 days post symptoms onset). The specificity of anti-S1 ELISA (IgG) is 96.7% (95% CI; 92.54-98.93) [41]. As our resources are limited, we do not have the means to perform the serum neutralization test, we decided to use the Abbott Architect technique which allows a quantitative determination of anti-RBD IgG antibodies, with an estimated sensitivity (based on > 14 days post-positive reverse transcription-PCR [RT-PCR] samples) and specificity were 98.3% (90.6% to 100%) and 99.5% (97.1% to 100%), respectively [42]. This method showed high concordance with the neutralizing antibody titers [43]. Please see Methods section in lines 114-116 and in lines 125-130.

Point 2: Please provide more details of the assays & their performance minimally from the package insert; preferably from published evaluations of these assays. Revise section 2.2 to make it clearer that all descriptions pertaining to assay (i) and (ii) is together.

Response 2: As highlighted by reviewer, details of the serological tests used in the study have been added in yellow in section 2.2. Please see methods section in lines 111-132: ‘’To detect IgG antibodies against SARS-CoV-2 spike protein in human serum, we used two serological techniques: semi quantitative enzyme-linked immunosorbent assay (ELISA) using the commercial kit (Anti-SARS-CoV-2 S1 ELISA IgG; Euroimmun, Lübeck, Germany) following the manufacture instructions [40].  This method has an estimated sensitivity of 75.0% (95% CI 47.62-92.73 (based on > 14 days post symptoms onset) and a specificity of 96.7% (95% CI; 92.54-98.93) [41]. The results were analyzed semi-quantitatively and interpreted according to the manufacturer instructions. We evaluated them by calculating a ratio of extinction of the control or patient sample to the extinction of the calibrator. This ratio is interpreted as follows: <0.8 negative; ≥0.8 to <1.0 limit; 1.1 positive. The borderline results were considered negative for the analysis. A quantitative immunoassay of chemiluminescent microparticles (CMIA) to detect antibodies against the receptor-binding domain (RBD) of the S1 subunit of the SARS-CoV-2 spike protein. The sequence used for the RBD was from the WH-Human 1 coronavirus.  The assay is performed on the Abbott Architect i2000SR instrument (Abbott Laboratories, Abbott Park, Illinois) following the manufacturer's instructions for the SARS-CoV-2 IgG II Quant [42]. This method has a sensitivity (based on > 14 days post-positive reverse transcription-PCR [RT-PCR] samples) and specificity were 98.3% (90.6% to 100%) and 99.5% (97.1% to 100%), respectively [42]. This method showed high concordance with the neutralizing antibody titers [43] and can identifies antibodies in patients with two variants of concern (VOC 202012/V1 [UK] and (VOC 202012/V2 [South Africa] stains [42]. The analytical measurement range defined by the manufacturer is 21 to 40,000 AU/ml and the cut-off point is ≥ 50 AU/ml.’’ 

Point 3: Since the IgG assays are semi-quantitative/quantitative expressing the results in concentration or magnitude of cut-off index will be more useful to compare between the 2 different vaccines rather than express results as percent positive or negative. 

Response 3: Thank you for your comment. We expressed the results qualitatively as percentages (Table 1 and 2) and quantitatively as antibody titers (Figure1).

Point 4: What is the evidence that your healthcare workers might not have been exposed to subclinical/asymptomatic CoVID19 in the 5 months after their vaccination? 

Response 4: Thank you for your interesting comment. At the beginning of the study, to ensure that health workers did not develop SARS-CoV-2 infection during the 5 months following the second dose of vaccine, we relied on the questionnaire where volunteers declared at the time of sampling that they had never been infected with SARS CoV2. In accordance with your recommendation, an Architect anti-N antibody test was performed on participants vaccinated with ChAdOx1 nCoV-19 vaccine and all participants was found to be negative except one. Therefore, we have modified the results by excluding the participant with anti-N antibodies from the study.  This modification did not change the result interpretation. Unfortunately, for individuals vaccinated by BBIBP-CorV, the measurement of anti-N antibodies cannot distinguish between infected and vaccinated individuals. We have added these sentences in lines 151-156” To check that vaccinated individuals are previously infected with SARS-CoV-2 infection, we assessed IgG antibodies against SARS-CoV-2 nucleocapsid protein using SARS-CoV-2 IgG test (Architect IgG test, Abbott, USA). The anti-N antibody test was performed in participants vaccinated with of ChAdOx1 nCoV-19 vaccine (42). All participants were found to be negative except one. The latter was excluded from the study (data not shown)

Point 5: Several of your references cited are incomplete without journal title, volume, page numbers or doi - 5, 16, 28, 29, 30, 33, 35, 37, 39, 43, 45, 46, 57, 64, especially the MedRxIV citations: 23, 26, 47, 48, 50, 58, 

Response 5: Thank you for your comment, we have modified the incomplete references as recommended. Since we added references after the correction, reference 43 became 47, 45 became 49, 46 became 50, 47 became 51, 48 became 52, 50 became 54, 57 became 61, 58 became 63 and 64 became 76.

Reviewer 2 Report

Comments on the manuscript: “Anti-SARS-CoV-2 antibody Responses 5 Months Post Complete Vaccination of Moroccan Healthcare Workers”. By Najlaa Assaid et al.

Authors collected sera from 83 Health Care Workers (HCW) 5 months after receiving the 2nd dose of AstraZeneca ChAdOx1 (n=42) vaccine or inactivated Sinopharm vaccine (n=41). They report HCW anti-SARS-CoV-2 IgG antibodies levels assessed using 2 test: Euroimmun ELISA and Abbott Architect. SARS-CoV-21 IgG test. Among 83 participants, 66.27% are seropositive for IgG using Euroimmune ELISA and 86.75% were positive with Abbott Architect. No age or gender statistically differences were found as well between antibody levels induced by the evaluated vaccines.

Manuscript is well-written and clearly explained, data supports author´s conclusions. Data is relevant to understand long-lasting antibody responses to vaccines. Most of the data available come from high-income countries and little is know from low-middle income countries were other ethnicities, epidemiological and vaccination conditions could have an impact on vaccines performance.

Major points:

  1. To better understand the epidemiological context of the study will be useful to have a graph showing the number of COVID-19 cases over the time and indicate there the dates of volunteer’s vaccination and the sampling period for this study.
  2. There is no information about breakthrough infections among the HCW cohort. In figure 1 is shown some volunteers with more than 20 times more antibody values. Measuring anti-N antibody titres could differentiate the infected (even asymptomatic).
  3. Is not clear if all HCW were not previously infected with SARS-CoV-2.

Is important to determine if volunteers had previous or post vaccination SARS-CoV-2 infection to strength results interpretations.

Author Response

Response to Reviewer 2 Comments

All modifications in the manuscript are highlighted in yellow.

Point 1: To better understand the epidemiological context of the study will be useful to have a graph showing the number of COVID-19 cases over the time and indicate there the dates of volunteer’s vaccination and the sampling period for this study.

Response 1: Thank you for your comment. We have added this paragraph to indicate the dates of volunteer vaccination and the sampling period for this study in lines 103-107 “Participants received the second dose of ChAdOx1 nCoV-19 or BBIBP-CorV vaccine at the same time (March 2021). After five months (August 2021), we collected samples to measure the level of anti-SARS-CoV-2 antibodies. Volunteers are reported who have never been infected with SARS-CoV2.”

Point 2: There is no information about breakthrough infections among the HCW cohort. In figure 1 is shown some volunteers with more than 20 times more antibody values. Measuring anti-N in lines antibody titres could differentiate the infected (even asymptomatic)

Response 2: Thank you for your interesting comment. At the beginning of the study, to ensure that health workers did not develop SARS-CoV-2 infection during the 5 months following the second dose of vaccine, we relied on the questionnaire where volunteers declared at the time of sampling that they had never been infected with SARS-CoV-2. In accordance with your recommendation, an Architect anti-N antibody test was performed on participants vaccinated with AstraZeneca vaccine and all participants was found to be negative except one. Therefore, we have modified the results by excluding the participant with anti-N antibodies from the study.  This modification did not change the result conclusion. Unfortunately, for individuals vaccinated with BBIBP-CorV vaccine, the measurement of anti-N antibodies cannot distinguish between infected and vaccinated individuals. We have added these sentences in lines 151-156”. To check that vaccinated individuals are previously infected with SARS-CoV-2 infection, we assessed IgG antibodies against SARS-CoV-2 nucleocapsid (N) protein using SARS-CoV-2 IgG test (Architect IgG test, Abbott, USA). The anti-N antibody test was performed in participants vaccinated with of ChAdOx1 nCoV-19 vaccine (42). All participants were found to be negative except one. The latter was excluded from the study (data not shown)

Point 3: Is not clear if all HCW were not previously infected with SARS-CoV-2. Is important to determine if volunteers had previous or post vaccination SARS-CoV-2 infection to strength results interpretations.

Response 3: Thank you for your advice to improve our manuscript. We have already responded to this comment in the previous comment.

Reviewer 3 Report

The manuscript investigated the persistence of the vaccine-induced immunoglobulin G (IgG) antibodies against SARS-CoV-2 five months after the second vaccination dose of either ChAdOx1 nCoV-19 or BBIBP-CorV among healthcare workers in Morocco. I think that the biological hypothesis addressed by this study is interesting and the experimental aim is in line with the objective of this work. Nonetheless, some points need to be addressed:

  • I think more details regarding the different antibody assays need to be included. Are the detected antibodies, against a specific SARS-CoV-2 variants? Standard curves should also be provided to compare the overall background and efficiency of each assay.

  • Graph in Figure 1 does not show the distributions of SARS-CoV-2 IgG antibodies before/after vaccination. The kinetic of Ig production in response to the two vaccines might indeed be different. Were all healthcare workers tested on day 0 (before the second dose of vaccine; T0) or at different time points? Please clarify.

  • The authors should expand their discussion and conclusions on the adequacy and efficiency of each vaccine compared to the others. Based on their data, there are no difference between the two vaccines. The authors should speculate more on this topic.

Author Response

Response to Reviewer 3 Comments

All modifications in the manuscript are highlighted in yellow.

Point 1 : I think more details regarding the different antibody assays need to be included. Are the detected antibodies, against a specific SARS-CoV-2 variants? Standard curves should also be provided to compare the overall background and efficiency of each assay.

Response 1 : Thank you for your comment. We have used commercial kits for both assays. Serological tests details used in the study have been added in yellow in section 2.2. Please see methods section in lines 111-132: ‘’To detect IgG antibodies against SARS-CoV-2 spike protein in human serum, we used two serological techniques: semi-quantitative enzyme-linked immunosorbent assay (ELISA) using the commercial kit (Anti-SARS-CoV-2 S1 ELISA IgG; Euroimmun, Lübeck, Germany) following the manufacture’s instruction’s [40].  This method has an estimated sensitivity of 75.0% (95% CI 47.62-92.73) (based on > 14 days post symptoms onset) and a specificity of 96.7% (95% CI; 92.54-98.93) [41]. The results were analyzed semi-quantitatively and interpreted according to the manufacturer’s instructions. We evaluated them by calculating a ratio of extinction of the control or patient sample to the extinction of the calibrator. This ratio is interpreted as follows: <0.8 negative; ≥0.8 to <1.0 limit; 1.1 positive. The borderline results were considered negative for the analysis. Quantitative immunoassay of chemiluminescent microparticles (CMIA) to detect antibodies against the receptor-binding domain (RBD) of the S1 subunit of the SARS-CoV-2 spike protein was used. The sequence used for the RBD was from the WH-Human 1 coronavirus.  The assay is performed on the Abbott Architect i2000SR instrument (Abbott Laboratories, Abbott Park, Illinois) following the manufacturer's instructions for the SARS-CoV-2 IgG II Quant [42]. This method has a sensitivity (based on > 14 days post-positive reverse transcription-PCR [RT-PCR] samples) and specificity were 98.3% (90.6% to 100%) and 99.5% (97.1% to 100%), respectively [42]. This method showed high concordance with the neutralizing antibody titers [43] and can identifies antibodies in patients with two variants of concern (VOC 202012/V1 [UK] and (VOC 202012/V2 [South Africa] stains [42]. The analytical measurement range defined by the manufacturer is 21 to 40,000 AU/ml and the cut-off point is ≥ 50 AU/ml.’’ 

Point 2 : Graph in Figure 1 does not show the distributions of SARS-CoV-2 IgG antibodies before/after vaccination. The kinetic of Ig production in response to the two vaccines might indeed be different. Were all healthcare workers tested on day 0 (before the second dose of vaccine; T0) or at different time points? Please clarify.

Response 2 : We agree that the kinetics of antibody production could be different between the two vaccines. However, the purpose of our study is to compare the persistence of antibodies after five months of the second dose of two vaccines: ChAdOx1 nCoV-19 and BBIBP-CorV. In our study, health care workers were not tested before the second dose but only one assay was performed five months later to assess the persistence of antibodies induced by vaccination. To clarify we have added five months after the second vaccine dose in the legend of figure 1.

Point 3 : The authors should expand their discussion and conclusions on the adequacy and efficiency of each vaccine compared to the others. Based on their data, there are no difference between the two vaccines. The authors should speculate more on this topic.

Response 3 : Thank you for your comment. To describe the efficacy of the vaccines used in our study, we have added the following paragraph to the lines 312-318:

“Several studies have shown the efficacy of the ChAdOx1 nCoV-19 vaccine in the prevention of symptomatic and severe cases of COVID-19 in different populations including the elderly (significant vaccine efficacy of 70.4% after two doses) [68–70]. Although, its efficacy represented by neutralizing antibody activity is much less when compared to mRNA vaccines [71]. Indeed, it was reported 90% efficacy for the BNT162b2 vaccine [72] and 94.5% for the mRNA-1273 vaccine [73]. An efficacy of 78.1% in healthy adults has been showed for the BBIBP-CorV inactivated vaccine [74].

We did not find significant differences between the median level of antibodies produced by the two vaccines used in the study probably due to sample size, we have added the following sentence in lines 306-309: “and this is probably due to the sample size. The cellular immune response may be different between the two vaccines, which was not evaluated in our study. Also, it would be interesting to compare the humoral response after vaccination at different time points.”

Round 2

Reviewer 1 Report

Having compared a semi-quantitative serology assay with a quantitative one with better sensitivity you should state in the discussion that the use of sensitive assays is preferred. 

You could cite de la Monte Clinical Pathology 2021; 14:1-9 that suggest a protective threshold for the Abbott assay as >4000 AU/mL.

Another article that appeared in this Journal Lau et al "Vaccines" 2021; 9:1241 reported a mean value of 1210 AU/mL using the same Abbott assay at 5 months post-vaccination.  

Reviewer 2 Report

Authors attend the points raised in my first review.

Author Response

No comment from Reviewer 2

Reviewer 3 Report

The authors addressed all my issues.

Author Response

No comment from Reviewer 3